# Validation of the human disharmony loop: Pectoralis minor tenotomy significantly reduces pain and improves function in historically challenging patients who meet reproducible and explicit diagnostic criteria

**James M. Friedman** [1]*, **Jaicharan Iyengar**[1], **Ketan Sharma**[2]

**1** Sutter Alpine Care Clinic, Stockton, California, United States of America, **2** St. Luke's Plastic and Reconstructive Surgery, Meridian, Idaho, United States of America

* James.Friedman46@gmail.com

## Abstract

### Background/Objectives

Patients commonly present with a mix of intractable shoulder pain, persistent impingement/loss of shoulder motion, neck pain, headaches, and distal neuropathy. These patients are notoriously resistant to surgical and non-surgical treatments. Previously we proposed the Human Disharmony Loop (HDL) as a model that anatomically explains these symptoms and diagnostically predicts successful response to surgical intervention. The purpose of this study is to validate positive surgical treatment outcomes in patients diagnosed in the HDL via retrospective chart review. We hypothesized that pectoralis minor release would reliably decrease pain and occipital headaches and increase shoulder motion for patients who met diagnostic criteria for the HDL.

### Methods

Patients diagnosed with the HDL and treated with pectoralis minor release at two separate institutions with at least 6-month follow-up were included. Diagnosis was based on explicit anatomic and symptomatic criteria: coracoid tenderness, scapular protraction, and at least one end symptom. Neuropathy was tested using the scratch-collapse test. Outcomes included pain scores, clinical neuropathic lesions, rotator cuff impingement signs, shoulder range of motion, and complications.

### Results

115 patients were included. Average age was 48. 37% were male. 89% of patients who received a preoperative subcoracoid injection reported a significant decrease

**Data availability statement:** All relevant data are within the manuscript and its Supporting Information files.

**Funding:** The author(s) received no specific funding for this work.

**Competing interests:** The authors have declared that no competing interests exist.

in presenting symptoms. 6 months after PM release, median VAS pain scores decreased from 8 to 2. Occipital headaches decreased from 66% to 6%. Rotator cuff impingement decreased from 87% to 10%. Median shoulder abduction increased from 90 to 180 degrees. Neuropathy decreased at the following locations: scalenes 57–2%, suprascapular 51–0%, quadrilateral 81–5%, radial 60–11%, cubital 31–25%, carpal 53–25%. 25% of patients required secondary distal neurolysis. Complications remained low at 3% (3 seroma, 1 wound dehiscence).

## Conclusions

Patients diagnosed with the Human Disharmony Loop exhibit a dramatic clinical improvement following pectoralis minor release. A medial coracoid pectoralis minor block injection can aid in diagnosis but does not rule-out the syndrome. Patients showed significant reductions in shoulder pain, headaches, concomitant neuropathic lesions and improved shoulder range of motion. Patients should be counseled that some may need secondary neurolysis for residual neuropathy.

## Introduction

Historically, evaluation and treatment of patients presenting with chronic and vague neck, upper back, and shoulder pain with radiating arm symptoms has been notoriously challenging. These patients often exhibit non-specific MRI findings of rotator cuff tendinopathy or degenerative labral pathology that does not explain their diffuse symptoms. Previous work investigating this common yet complex shoulder pathology includes but is not limited to: neurogenic pectoralis minor (PM) syndrome [1], upper crossed syndrome [2], scapulothoracic abnormal motion [3], dorsal scapular nerve neuropathy [4], neurogenic thoracic outlet syndrome [5], pectoralis minor syndrome [6], SICK scapula syndrome [7]. However, each of these models suffer from vague diagnostic criteria that is difficult to reproduce clinically. Because of this difficulty in diagnosis, effective treatment for these patients has also been difficult to measure. Pectoralis minor release has gained increasing attention and is associated with good clinical outcomes with minimal long-term complications in appropriately identified patients. [3,5,6,8,9] However, correct prediction of patients who will most benefit from PM release has been limited by a lack of diagnostic framework and a lack of anatomic understanding behind the symptom physiology. In a previous paper we introduced, via a small case series, a novel model of upper extremity pain, titled The Human Disharmony Loop (HDL), to more easily diagnose, explain, and treat patients with historically challenging upper extremity pain. [10]

The HDL focuses on the central role of the scapula as the connection between the axial skeleton and upper extremity. The scapula is balanced by the posterior musculature innervated by the upper brachial plexus (C5, C6) versus the anterior PM innervated by the lower brachial plexus (C8-T1). In the HDL, the asymmetric lower trunk innervation to the PM causes scapular dyskinesia. This scapular dyskinesia

generates traction on the upper brachial plexus leading to weakness of the posterior stabilizers while leaving the PM unaffected. PM tightness therefore reinforces itself via a central positive feedback loop. (Fig 1) The scapular dyskinesia also causes stretch of attached muscles, compression of the subacromial and costoclavicular spaces, and traction to the brachial plexus. The resultant symptoms include occipital headaches, peri-scapular pain and tightness involving the trapezius and rhomboids, shoulder weakness with overhead reach and rotator cuff impingement, forearm burning/throbbing, and hand numbness/tingling and weakness. (Fig 2) With this model anatomically describing the symptom physiology, we developed explicit and reproducible criteria to diagnose patients with the HDL. The diagnostic criteria are coracoid tenderness, scapular dyskinesia (specifically protraction), and at least one of the resultant end symptoms: occipital headaches or neck tightness, peri-scapular muscle pain, rotator cuff impingement, and/or proximal/distal neuropathy. (Table 1)

Sanders and Annest previously described a PM muscle block to diagnose Pectoralis Minor Syndrome (1) which shares some of the symptoms of HDL, although it does not include scapular dyskinesia or any resultant scapular pathology. In

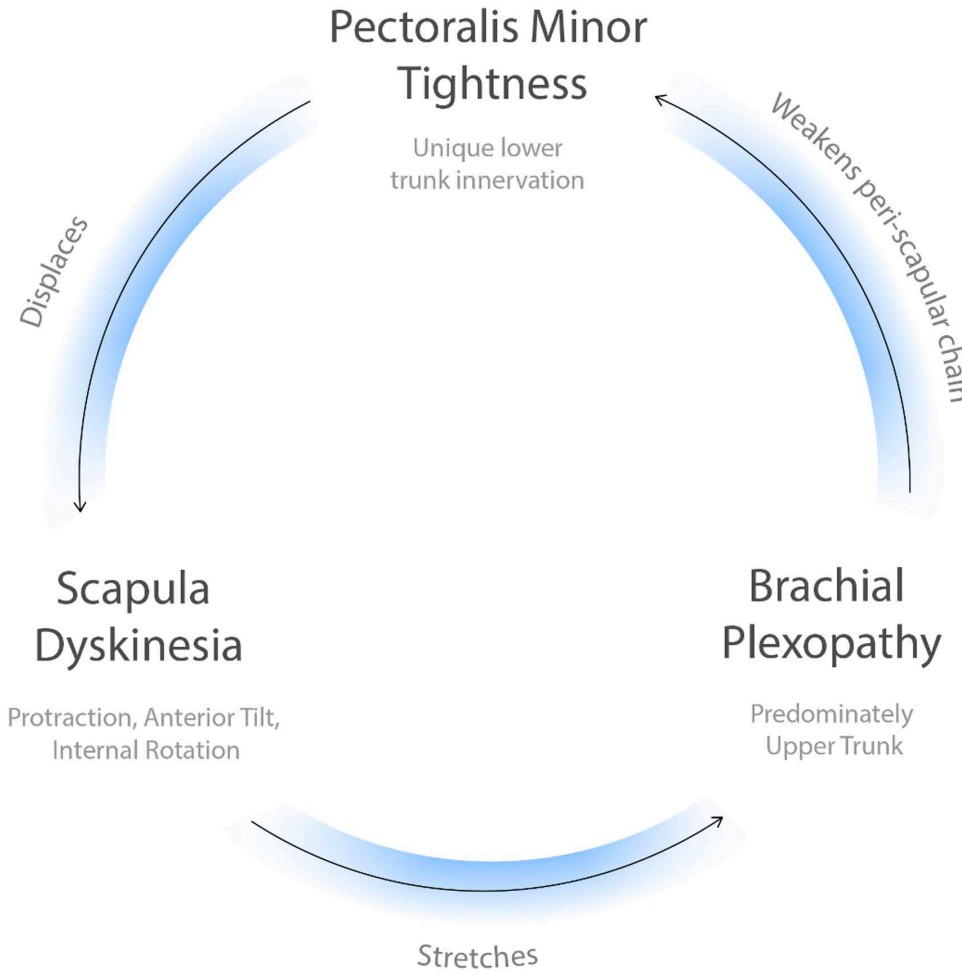

**Fig 1. The Human Disharmony Loop (central).** The central positive feedback loop is visualized: pectoralis minor tightness protracts the scapula which stretches the brachial plexus primarily the upper trunk which therefore reinforces pectoralis minor tightness due to its unique lower trunk C8-T1 innervation. The positive feedback nature of the loop produces the intractable and resistant symptomology of chronic pain, neuropathy, and weakness of the human upper limb.

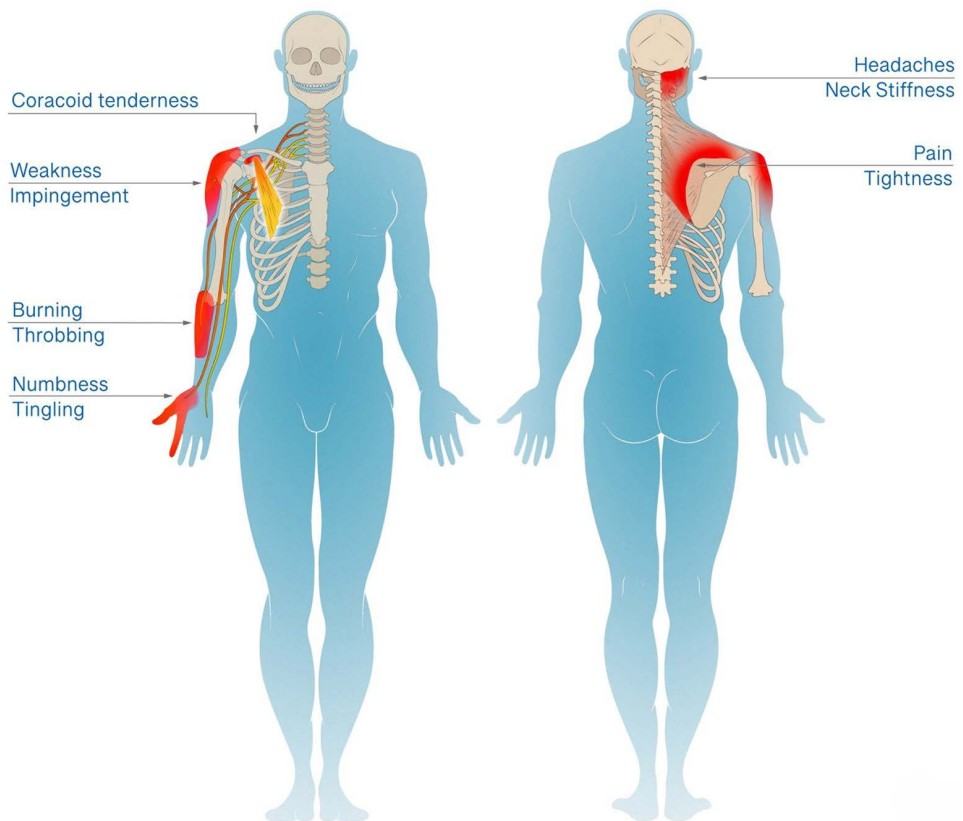

**Fig 2. The Human Disharmony Loop Clinical Presentation.** A single anatomic source, the tight PM disturbing the scapula, produces coracoid tenderness with five terminal clusters of symptoms: (1) occipital headaches and neck stiffness, (2) peri-scapular pain and tightness, (3) shoulder pain and weakness with overhead reach, (4) forearm burning/throbbing, and (5) hand weakness and numbness/tingling.

**Table 1. Diagnostic Criteria for Human Disharmony Loop.**

| Anatomic | 1. Coracoid tenderness + |
|---|---|
| | 2. Scapular dyskinesia (protraction at rest or with overhead reach) + |
| Symptomatic | 3. At least one or more of:<br> a. Occipital headaches or neck stiffness<br> b. Peri-scapular pain and tightness (upper trapezius and rhomboids trigger points or muscle knots)<br> c. Shoulder subacromial pain and weakness (positive impingement signs and limited ROM)<br> d. Radiating neuropathy (burning/throbbing/numbness/tingling/weakness) to forearm and hand |

Crucially, Human Disharmony Loop is a syndrome where patients must meet all three strict anatomic and symptomatic criteria.

summary, an injection of lidocaine is given just distal to the PM insertion at the coracoid. A reduction in pain was reported as a strong predictor for PM involvement. We also performed PM muscle blocks to assess efficacy for diagnosis of HDL.

The potential value of the HDL is twofold. First, the model anatomically explains historically vague and challenging symptoms associated with upper extremity chronic pain. Second, the HDL provides explicit and reproducible diagnostic

criteria allowing for hypothesis-driven testing of treatment. If our model and diagnostic criteria are correct, we would expect a PM release to reliably 'break the loop', restore proper scapula kinesis, and result in substantial clinical improvement. The purpose of this paper is to validate the applicability of HDL diagnosis via a large retrospective chart review of patients treated using our proposed diagnostic criteria. We hypothesize that patients who met HDL diagnostic criteria would show a positive clinical response to PM release in terms of decreased pain, decreased occipital headaches, and increased shoulder motion.

## Methods

This is a retrospective case series of patients from two separate clinics. All were treated by a fellowship-trained board-certified hand or sports surgeon. Inclusion criteria included: age > 12 years; physical exam meeting the criteria of the HDL (Table 1); persistent symptoms following at least 2 months of targeted physical therapy. Exclusion criteria included: follow-up < 6 months; unsafe surgical candidates. Patients with concomitant rotator cuff tears underwent surgical repair separately prior to any treatment for HDL. Patients were offered a preoperative PM block(1), with pain relief or improvement in range of motion noted subsequently. Patients were evaluated pre-operatively, and at 2, 6, 12, and 24 weeks. At each visit, patients completed self-reported pain diagrams and symptoms questionnaires. (Fig 3) Upper extremity

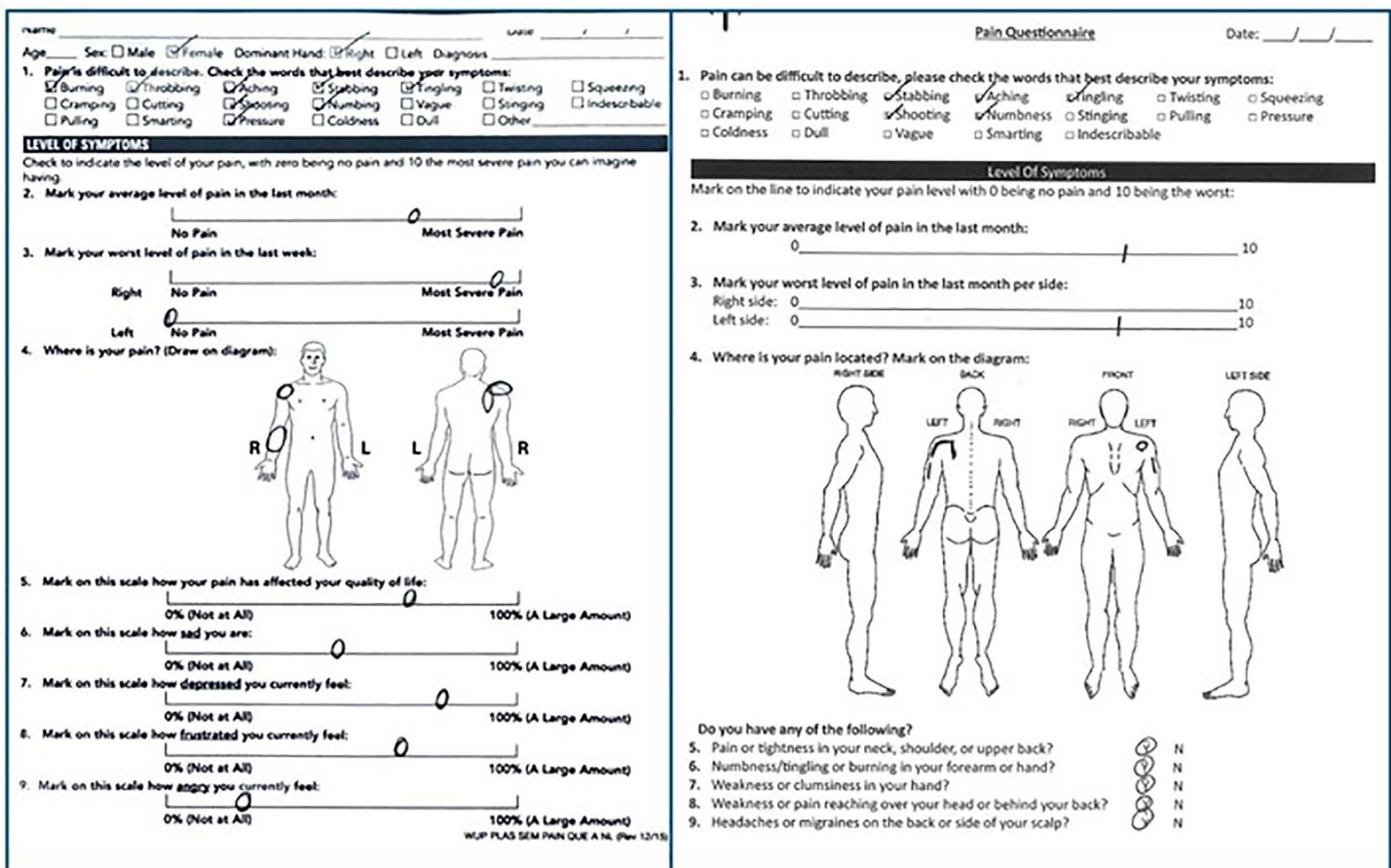

**Fig 3. Sample Pain Diagrams.** Two sample pain diagrams from unrelated patients. Pain is global but tends to distribute along anatomic lines: coracoid tenderness, peri-scapular pain, radiating neuropathy.

neuropathic lesions were identified using the scratch collapse test [11] at the following locations: scalene muscles, pectoralis minor, suprascapular notch, quadrilateral space, and radial, cubital, and carpal tunnels. Musculoskeletal exams included Medical Research Council (MRC) grade muscle testing and shoulder range of motion (ROM) values in abduction plane. Scapula dyskinesia was subclassified into four stages: I or none (no protraction), II or dynamic (protraction with overhead reach only), III or static reversible (protraction at rest but manually reversible via the examiner), IV or static irreversible (protraction at rest not manually reversible via the examiner). Inciting factors that introduced patients into the loop were recorded. (Fig 4) Continuous and categorical variables of interest were compared using Student's $t$-test and chi-squared analysis, respectively, using STATA v19.0.

Prior to surgical consideration, all patients underwent at least two months of physical therapy focused on anterior chain stretching and posterior chain strengthening. Patients who failed non-operative treatment were treated with open PM release with infraclavicular brachial plexus neurolysis, followed by physical therapy emphasizing upper trapezius and rhomboid strengthening. PM tenotomy was achieved via a previously-described open approach. [10] Patient variables included general demographics and workers compensation status. Outcomes included self-reported pain scores, scapular dyskinesia stage, rotator cuff impingement signs, shoulder range of motion, presence of headaches in the occipital scalp, presence of neuropathic lesions, surgical complications. Ethics board approval was obtained, and need for consent was waived by the ethics committee as all data was anonymous, the study was observational only, and the study posed minimal risk to the included patients. Data collection occurred on 5/1/25. Data was collected by patient's respective treating physician and combined into an anonymous database that did not contain patient identifying information. Authors did not have access to patient identifying data during or after data analysis. The study adhered to the STROBE guidelines.

## Results

$N = 115$ patients were included. Median age was 48. Sex was 37% male and 63% female. Median BMI was 29. 26% were workers compensation. Inciting factors entering the HDL were: female athlete (4%), breast cancer (3%), macromastia (18%), postoperative reverse total shoulder arthroplasty (rTSA) (11%), cervical spine disease (12%), history of trauma (25%), bodybuilder (4%), iatrogenic upper trunk regional anesthetic injury (3%), plexus traction injury (5%), overhead athlete (2%), and idiopathic (8%). (Table 2)

At presentation, median pre-operative pain was 8/10. Scapular dyskinesia on exam was categorized as stage 2 (0%), stage 3 (52%), stage 4 (43%). On exam, 87% of patients exhibited rotator cuff impingement, and median shoulder abduction ROM was 90 degrees. 66% of patients endorsed headaches in the occipital region. Prevalence of preoperative concomitant neuropathic lesions was scalene muscles (57%), suprascapular notch (51%), quadrilateral space (81%), radial tunnel (60%), cubital tunnel (31%), and carpal tunnel (53%). 87 patients received a pre-operative PM block with 77 patients (89%) endorsing significant improvement in presenting symptoms. Baseline severity of disease correlated strongly with increasing stage of scapular dyskinesia, except for cubital tunnel. (Table 3)

Six months following surgery, median pain was 2/10. Post-operative scapular dyskinesia redistributed to stage I (95%), stage II (4%), stage III (0%), stage IV (0%). Only 10% of patients exhibited continued rotator cuff impingement, and median shoulder abduction ROM increased to 180 degrees. Only 6% of patients endorsed occipital headaches. Prevalence of concomitant neuropathic lesions decreased to scalene muscles (2%), suprascapular notch (0%), quadrilateral space (5%), radial tunnel (11%), cubital tunnel (25%), carpal tunnel (25%). All differences were statistically significant ($p < 0.01$). (Fig 5) There were 4 complications (3%): 3 seromas that resolved spontaneously, and 1 wound dehiscence requiring operative takeback for closure. 29 patients (25%) underwent a secondary neurolysis for persistent neuropathic symptoms. (Table 4)

## Discussion

In this paper we analyzed the outcomes of PM release on patients who met diagnostic criteria for the Human Disharmony Loop. Our findings support our hypothesis that patients in the HDL will benefit substantially showing improvement of pain,

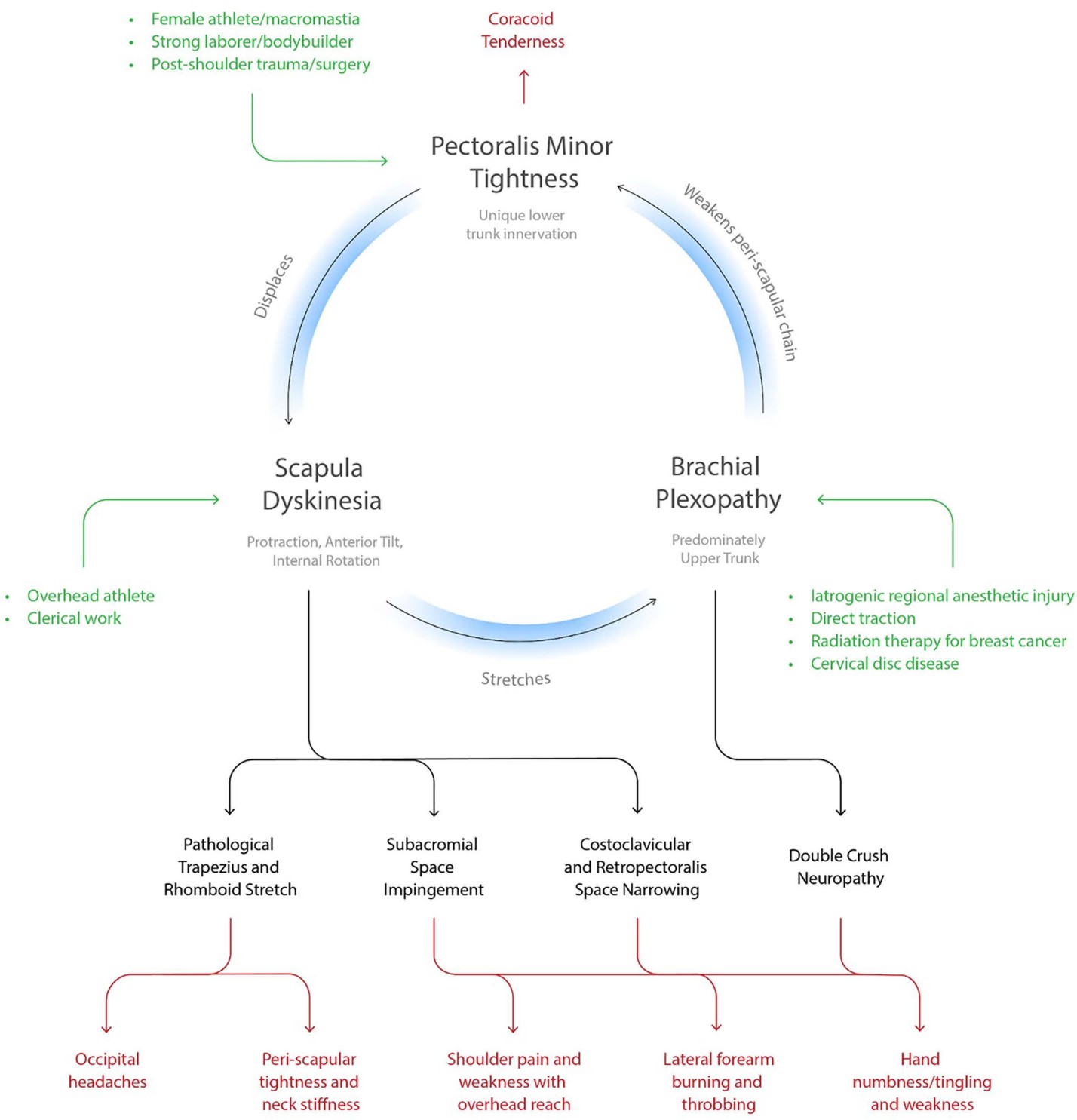

**Fig 4. The Human Disharmony Loop (full) Including Inciting Factors.** The central loop has three elements, each causing anatomic sequelae. Diverse groups of patients can enter via each element, seen in green. The anatomic sequelae then produce five distinct groups of clinical symptoms (bottom row).

**Table 2. Patient Demographics.**

| Variable | Number |
|---|---|
| Total | 115 |
| Age (median) | 48 |
| Male | 43 (37%) |
| BMI (median) | 29 |
| Workers Compensation | 30 (26%) |
| Inciting Factors | |
| Female Athlete | 4 (4%) |
| Breast Cancer | 3 (3%) |
| Macromastia | 21 (18%) |
| Postoperative reverse TSA | 13 (11%) |
| Cervical Spine Disease | 14 (12%) |
| History of Trauma | 29 (25%) |
| Bodybuilder | 5 (4%) |
| Upper Trunk Block | 3 (3%) |
| Plexus Traction | 6 (5%) |
| Overhead Athlete | 2 (2%) |
| Idiopathic | 9 (8%) |
| Laterality | Right (57%) |
| Hand Dominance | Right (81%) |

**Table 3. Baseline Severity of Disease by Scapular Dyskinesia Stage[1].**

| Symptom | Scapular Dyskinesia Stage 3 | Scapular Dyskinesia Stage 4 | P-Value |
|---|---|---|---|
| Pain (average) | 6.9 | 8.2 | <0.01 |
| Range of Motion (average, degrees) | 116 | 89 | <0.01 |
| Occipital Headaches (incidence) | 53% | 84% | <0.01 |
| Concomitant Neuropathy[2] (incidence) | | | |
| Scalene muscles | 50% | 76% | <0.01 |
| Suprascapular notch | 45% | 67% | <0.01 |
| Quadrilateral space | 80% | 96% | <0.01 |
| Radial tunnel | 54% | 78% | <0.01 |
| Cubital tunnel | 32% | 31% | 0.96 |
| Carpal tunnel | 47% | 65% | <0.01 |

[1]Scapular dyskinesia was staged on exam as: I (no protraction); II dynamic (protraction with overhead reach only); III static reversible (protraction at rest but manually reversible by examiner); IV static irreversible (protraction at rest but not manually reversible).

[2]Neuropathic lesions were considered positive with a positive scratch-collapse test at each anatomic location.

mechanical, and neuropathic symptoms following PM release and infraclavicular neurolysis. Of note, our findings were consistent across a diverse and historically challenging patient population including worker's compensation. [12] To our knowledge this is the first paper to validate a reproducible diagnostic approach to upper extremity dysfunction that predicts a successful treatment with PM release.

The HDL is a unique model of shoulder and upper limb pain that provides a clear anatomic explanation for the diversity and ubiquity of presenting symptoms and answers why they respond to PM release. In our model, the driver of all symptoms stems from scapular dyskinesia caused by a relatively tight PM, which due to its unique lower trunk innervation,

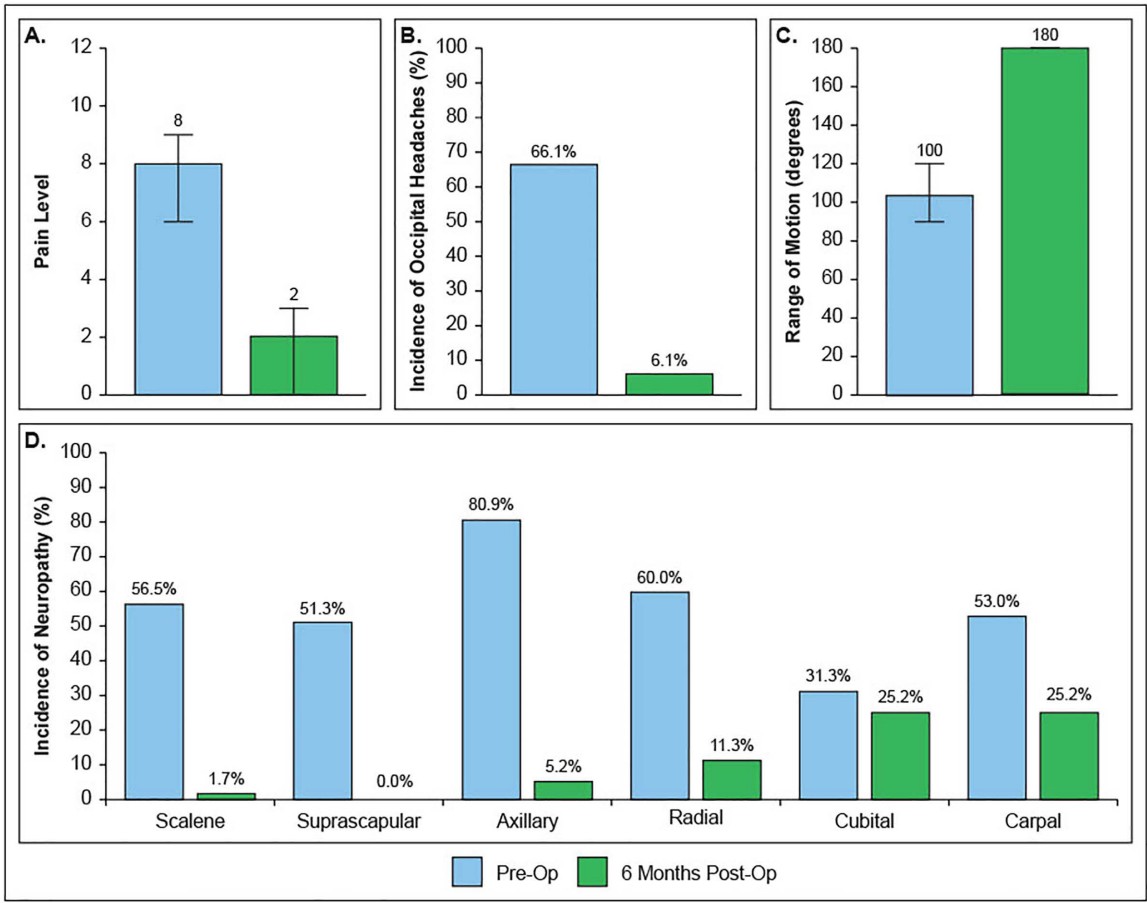

**Fig 5. Pain and Function Outcomes Following PM Tenotomy.** A. Pain level as represented by VAS pain scores. B. Percentage of patients reporting occipital headaches. C. Range of shoulder abduction. D. Percentage of patients presenting with positive scratch collapse testing common compression sites. Error bars represent interquartile range.

overpowers the dorsal scapula stabilizing muscles and impairs scapulothoracic motion. Given the central role of the scapula in coordinating function between the body and upper limb, this derangement then produces numerous downstream effects: narrowing of the costoclavicular and subacromial spaces, pathological stretch to the upper trapezius and rhomboids, irritation of the scalp occipital nerves, and traction to the brachial plexus. (Fig 6) This differs from previous shoulder models which report PM tightness as secondary to scapular dyskinesia [7], or unrelated to scapular dyskinesia. [1,13] We observed that PM release resolved scapular dyskinesia in almost all patients, which has been noted before albeit in a more limited patient population. [8] This lends strong evidence to our prediction that scapular dyskinesia is primarily caused by the asymmetric neurologic innervation to the PM.

The significance of the HDL diagnostic criteria and, by extension the HDL model, is further validated by the dramatic reduction in pain and improvement in function following PM release and infraclavicular neurolysis. The minimum clinically important difference (MCID) for the VAS pain score in the upper extremity ranges from 1.4–2. [14] In contrast, we found a median VAS pain score difference of 6 points or threefold higher, implicating the PM and its disturbance of the scapula as the true anatomic source. This is corroborated by baseline pain, impingement, and neuropathy all worsening with increasing stage of scapular dyskinesia, except for cubital tunnel. (Table 4) Furthermore, the HDL can account for all presenting

**Table 4. Pre-operative versus post-operative symptoms.**

| Symptom | Preoperative | Postoperative | P-Value |
|---|---|---|---|
| VAS pain score (median) | 8 | 2 | <0.01 |
| Scapular Dyskinesia Stage | | | |
| Stage I | 0 (0%) | 109 (95%) | |
| Stage II | 0 (0%) | 4 (4%) | <0.01 |
| Stage III | 60 (52%) | 0 (0%) | |
| Stage IV | 49 (43%) | 0 (0%) | |
| Mechanical Symptoms | | | |
| Rotator cuff impingement | 100 (87%) | 12 (10%) | |
| Shoulder abduction ROM (median) | 90 degrees | 180 degrees | <0.01 |
| Occipital headaches | 76 (66%) | 7 (6%) | |
| Neuropathic Lesions[1] | | | |
| Scalene muscles | 65 (57%) | 2 (2%) | |
| Suprascapular notch | 59 (51%) | 0 (0%) | |
| Quadrilateral space | 93 (81%) | 6 (5%) | <0.01 |
| Radial tunnel | 69 (60%) | 13 (11%) | |
| Cubital tunnel | 36 (31%) | 29 (25%) | |
| Carpal tunnel | 61 (53%) | 29 (25%) | |
| Secondary Neurolysis[2] | | 29 (25%) | |
| Scalene | | 1 (1%) | |
| Suprascapular | | 0 (0%) | |
| Quadrilateral space | N/A | 8 (7%) | N/A |
| Radial | | 16 (14%) | |
| Cubital | | 18 (16%) | |
| Carpal | | 11 (10%) | |

Symptoms are categorized into overall pain, scapular dyskinesia, mechanical symptoms, and neuropathic lesions.

[1]Neuropathic lesions were considered positive with a positive scratch-collapse test at each anatomic location.

[2]Despite significant reduction in symptoms following PM release, some patients suffered from persistent neuropathy and required secondary neurolysis.

symptoms including shoulder weakness, headaches, and upper back and neck pain, as opposed to models such as Pectoralis Minor Syndrome [6] which ascribe symptoms to infraclavicular brachial plexus compression. [1,12] However, a compressive lesion at the PM insertion would only account for distal neuropathic symptoms and cannot explain shoulder impingement or proximal pain such as headaches. Symptoms of the HDL can be stratified into muscle-related pain, shoulder mechanical symptoms, and neuropathic pain. (Fig 2) Muscle-related pain includes muscle pain of the rhomboid, muscle pain of the trapezius, and occipital headaches. As the PM tightens it depresses, anteriorly tilts, and internally rotates the scapula, collectively known as protraction. This displacement produces a chronic stretch on the trapezius and rhomboids. Eccentric muscle stretch is well known to cause damage and pain. [15] Furthermore, headaches evolve secondary to occipital nerve irritation or compression when passing through thickened trapezius fascia. [16] In the HDL model, stretching of the trapezius alters the fascia around the occipital nerve leading to occipital nerve neuritis and headaches. Restoration of scapula position restores trapezius and rhomboid muscle length tension relieving muscle pain and occipital nerve irritation. Our patients experienced dramatic reductions in muscle pain and occipital headaches with correction of scapular position following PM release. (Fig 5)

Mechanical symptoms of the HDL include rotator cuff impingement and associated loss of shoulder range of motion. It is generally accepted that scapular dyskinesia can compress the subacromial space causing rotator cuff impingement and

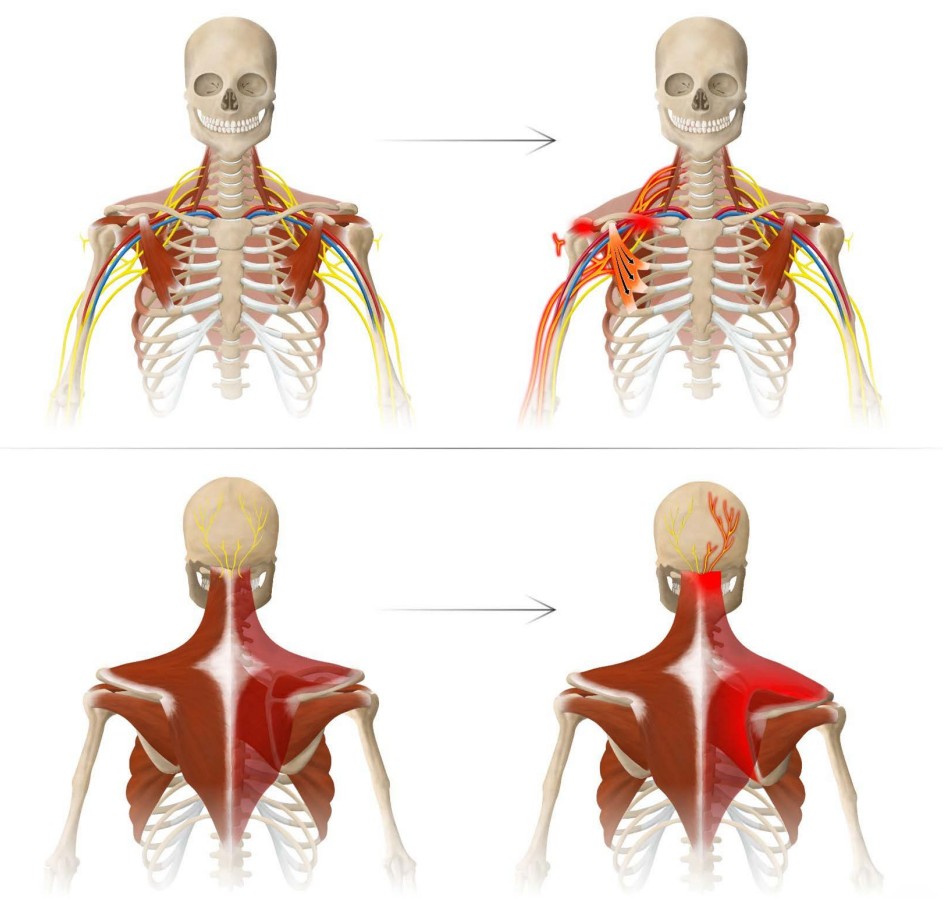

**Fig 6. Normal Anatomy vs. Disharmonic Pathoanatomy.** Top row: PM tightness stretches the upper brachial plexus roots and narrows the costo-clavicular and subacromial spaces, producing: (1) secondary neuropathy at the suprascapular notch, quadrilateral space, radial and carpal tunnels, (2) subacromial impingement, (3) thoracic outlet narrowing. Bottom row: PM tightness protracts the scapula which (1) stretches the upper trapezius and rhomboids and (2) irritates the occipital nerves to the scalp.

weakness. [9] In the setting of scapular dyskinesia, impingement symptoms have been shown to improve with scapular reposition and the assumed restoration of the subacromial space following PM release. [8] Our patients displayed significant restoration of shoulder ROM with dramatic reductions in impingement following PM release, supporting the hypothesis that these symptoms stem from PM tightness disturbing scapular mechanics. (Fig 5)

Neuropathic symptoms of the HDL include neurogenic pain along the radial forearm and entire hand as well as positive neuralgia testing at the scalenes, suprascapular space, quadrilateral space, pectoralis minor, radial tunnel, cubital tunnel, and/or carpal tunnel. Our study found near-complete reduction in neuropathic pain as well as reduction of positive scratch-collapse testing at the scalenes, suprascapular notch and quadrilateral space, with significant reduction at the radial and carpal tunnels, and less so at the cubital tunnel. This is consistent with our hypothesis that scapula displacement in the HDL stretches the brachial plexus but primarily affects the upper trunk more than the lower trunk, as the higher roots remain more prone to stretch and injury. [10] This is consistent with other known traction nerve pathologies such as Erb's palsy and burners experienced by contact athletes. Furthermore, the malposition of the scapula alters the soft tissue of the proximal upper limb girdle creating compression sites at the thoracic outlet, suprascapular notch, and quadrilateral space. The malposition of the scapula also lowers the clavicle inferiorly which narrows the costoclavicular

space and implicates the lower trunk innervation, including ulnar neuropathy. We believe that the more distal neurologic symptoms, including radial, carpal, and cubital tunnel, are likely a product of double crush [17], whereas the proximal neuropathic lesions likely originate from altered girdle anatomy. The HDL model predicts relief of neuropathic symptoms with restoration of the scapula position following PM tenotomy, but with possible residual symptoms in lingering areas of distal compression. This prediction was found true in this study with near-obliteration of proximal neuropathy but higher rates of persistent distal lesions following PM release. In this series, 25% of patients required secondary neurolysis for full relief, further validating the central role of the pectoralis minor in this double crush phenomenon. (Fig 7) Because we cannot yet predict which patients will require a secondary release, our current protocol is to release the PM first with close follow-up to assess for need of secondary releases.

The diagnosis of HDL is based on physical exam. Although MRI may help rule out other pathology in the shoulder, even common concomitant findings such as rotator cuff tears or labral tears do not explain the diversity of presenting symptoms and are not helpful diagnostically. EMG is another common test used to confirm focal demyelination of nerve; however, a negative EMG cannot be used to rule out the HDL and a single positive EMG focal finding similarly does not explain the diversity of symptoms experienced. In this study, we did find that 90% of patients who met HDL criteria responded to a targeted PM block. This test can therefore be useful, but it is important to note that a negative block test does not rule out the HDL, as the 10% who did not respond to the injection nonetheless responded similarly clinically to PM release. The rate of nonresponse to the injection could be due to technical error and/or severe PM tightness; this offers an avenue for future research.

This study has important limitations. First, as a case series, this study did not include a control group. It is therefore theoretically possible that patients would have improved without intervention or that outcomes were influenced by the placebo effect. However, all included patients did not respond to at least months of conservative management, our sample size is large, and these patients historically are resistant to treatment. We therefore think it is unlikely that patients would have

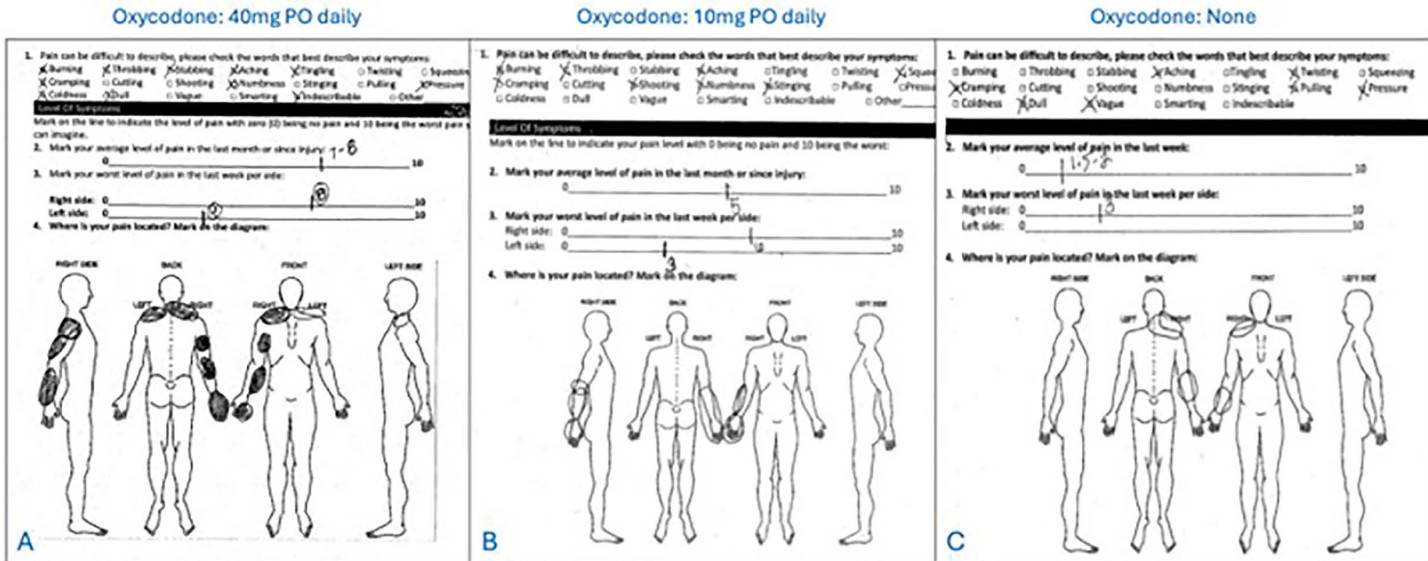

**Fig 7. Patient Example.** 34 y/o RHD M who fell off a roof several years prior and underwent ORIF of a humerus fracture. He had severe neck, upper back, shoulder, and arm pain precluding return to work and had been diagnosed with fibromyalgia. A: Pain diagram at presentation. B: Pain diagram 3 months following Right pectoralis minor tenotomy. C: Pain diagram 3 months following quadrilateral space decompression and radial and cubital tunnel release. He was able to return to work and transition off oxycodone for the first time in years. Breaking the loop is the first step to improvement, but surveillance and treatment for residual pain generators (in this case, axillary, radial, and ulnar neuropathy) is also crucial.

improved without surgical intervention. Future studies can include control groups to confirm this studies' findings. Second, our study only extends to six months, and patients may regress over time. We continue to monitor these patients longitudinally and intend to further validate or refute with longer, larger series. Third, we used the scratch-collapse test to identify focal neuropathies; however, this test is not included in formal diagnostic criteria for common compressive neuropathies. It is therefore important to stress that although we can claim that we observed improvement in distal neurologic symptoms as measured by scratch-collapse testing, we cannot definitively claim that PM release cured compressive nerve lesions. Fourth, this study does not measure the percentage of people who met diagnostic criteria for the HDL but responded to non-operative treatment. We maintain that all patients who meet HDL criteria should trial conservative physical therapy focused on anterior shoulder stretching before being considered for PM release. Finally, it is important to note that the HDL model is a predictive model based on clinical and anatomic observation. Although this study supports many aspects of the HDL model, this study does not definitively prove that the HDL model is complete. We expect that future studies will allow for continued refinement of the HDL.

## Conclusions

Patients with the Human Disharmony Loop present with some combination of occipital headaches, peri-scapular tightness, shoulder weakness, and distal radiating neuropathy. Patients who meet explicit and reproducible HDL anatomic and symptomatic diagnostic criteria respond reliably to pectoralis minor release showing significant improvements in pain, motion, and headaches. It should be noted, however, that a significant number of patients required secondary distal nerve releases, therefore patients should be counseled that a subsequent procedure to address "double crush" neuropathy may be indicated.

## Supporting information

**S1 File. This is the raw data used for this study.**
(XLSX)

## Author contributions

**Conceptualization:** James Friedman.

**Data curation:** James Friedman, Ketan Sharma.

**Formal analysis:** Ketan Sharma.

**Investigation:** Ketan Sharma.

**Methodology:** James Friedman, Jaicharan Iyengar, Ketan Sharma.

**Supervision:** Ketan Sharma.

**Validation:** Ketan Sharma.

**Visualization:** Ketan Sharma.

**Writing – original draft:** James Friedman.

**Writing – review & editing:** James Friedman, Jaicharan Iyengar, Ketan Sharma.

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
