## [Decision Letter · Decision Letter 0]

4 Aug 2025

Dear Dr. Friedman,

Thank you for submitting your manuscript to PLOS ONE. After careful consideration, we feel that it has merit but does not fully meet PLOS ONE’s publication criteria as it currently stands. Therefore, we invite you to submit a revised version of the manuscript that addresses the points raised during the review process.

Congratulations on your engaging and well-written paper on the “Human Disharmony Loop.”

Please address all reviewer comments, focusing on the following:

Clarify that the study is retrospective and update the abstract’s purpose accordingly.State clearly that the absence of a control group is a major limitation.Specify which statistical tests were used for each comparison.Indicate the patient enrolment period and the time from symptom onset to surgery.Provide more details on parameter changes at each follow-up point.

We look forward to receiving your revised manuscript.

Kind regards,

Emil George Haritinian, M.D, Ph.D.

Academic Editor

PLOS ONE

Journal Requirements:

2. Please amend your list of authors on the manuscript to ensure that each author is linked to an affiliation. Authors’ affiliations should reflect the institution where the work was done (if authors moved subsequently, you can also list the new affiliation stating “current affiliation:….” as necessary).

4. Please remove all personal information, ensure that the data shared are in accordance with participant consent, and re-upload a fully anonymized data set.

Additional guidance on preparing raw data for publication can be found in our Data Policy (https://journals.plos.org/plosone/s/data-availability#loc-human-research-participant-data-and-other-sensitive-data) and in the following article: http://www.bmj.com/content/340/bmj.c181.long .

Additional Editor Comments:

Congratulations on your engaging and well-written paper on the “Human Disharmony Loop.”

Please address all reviewer comments, focusing on the following:

- Clarify that the study is retrospective and update the abstract’s purpose accordingly.

- State clearly that the absence of a control group is a major limitation.

- Specify which statistical tests were used for each comparison.

- Indicate the patient enrolment period and the time from symptom onset to surgery.

- Provide more details on parameter changes at each follow-up point.

Reviewers' comments:

Reviewer's Responses to Questions

**Comments to the Author**

1. Is the manuscript technically sound, and do the data support the conclusions?

Reviewer #1: Yes

Reviewer #2: Partly

Reviewer #3: Yes

2. Has the statistical analysis been performed appropriately and rigorously?

Reviewer #1: Yes

Reviewer #2: Yes

Reviewer #3: Yes

3. Have the authors made all data underlying the findings in their manuscript fully available?

Reviewer #1: Yes

Reviewer #2: Yes

Reviewer #3: Yes

4. Is the manuscript presented in an intelligible fashion and written in standard English?

Reviewer #1: Yes

Reviewer #2: Yes

Reviewer #3: Yes

Reviewer #1: First, I would like to thank you for the opportunity to review the article "Validation of The Human Disharmony Loop: Pectoralis minor tenotomy significantly reduces pain and improves function in historically challenging patients who meet reproducible and explicit diagnostic criteria."

I confirm that the article is clearly written, with logical conclusions.

Therefore, I accept the publication of this article.

Sincerely,

Dr. João Paulo Barile

Reviewer #2: The authors present an interesting and thought-provoking manuscript on the "Human Disharmony Loop" (HDL), a novel conceptual framework for a challenging patient population with chronic upper extremity pain. The proposed diagnostic criteria are explicit, and the reported outcomes following pectoralis minor (PM) tenotomy are impressive. The paper is well-written and the model is clearly explained. However, there are several major concerns that need to be addressed before this manuscript can be considered for publication.

Major Concerns:

Study Design and Inconsistency: The most critical issue is a major inconsistency regarding the study design. The abstract states the purpose is to "validate this model prospectively" , whereas the Methods section and main body repeatedly describe the study as a "retrospective chart review". This is a fundamental contradiction that must be corrected throughout the manuscript. The entire framing of the paper depends on it.

Lack of a Control Group: As a retrospective case series, the study lacks a control group. This is a significant limitation. The dramatic improvements could be influenced by the placebo effect, which is known to be substantial in surgical interventions for pain. The authors should more thoroughly discuss this limitation in the Discussion section and acknowledge how it might temper the interpretation of the results. Without a comparison group (e.g., patients meeting HDL criteria treated with conservative therapy alone for the same duration), it is difficult to definitively attribute the outcomes solely to the PM release.

Reliance on the Scratch-Collapse Test: The diagnosis of multiple neuropathic lesions relies on the scratch-collapse test. As the authors rightly note in their limitations, this test is not part of the formal diagnostic criteria for these conditions. While useful as a clinical tool, its reliability and specificity can be debated. This reliance weakens the claims regarding the resolution of specific nerve compressions (e.g., carpal, cubital tunnel). The authors should further contextualize these findings and be more cautious in their conclusions about "curing" these neuropathies.

The "Double Crush" Phenomenon and Secondary Surgery: The finding that 25% of patients required a secondary neurolysis is clinically very important and adds credibility to the report. However, the discussion could explore this further. Does this high rate of secondary surgery suggest that the HDL model, while valuable, may be incomplete? Or does it position PM release as simply the

first critical step in a multi-stage treatment for a more complex regional pathology? Expanding on this would add significant depth to the discussion.

Minor Concerns:

Statistical Methods Description: In the Methods section, please specify the exact statistical tests used to generate the p-values (e.g., Wilcoxon signed-rank test for pre- vs. post-operative median scores, Chi-squared or Fisher's exact test for categorical variables).

Role of Preoperative PM Block: The data shows that 89% of patients who received a PM block reported improvement. This implies 11% had a "negative" block but still proceeded to surgery and, presumably, had good outcomes. The discussion could briefly touch on the potential reasons for a false-negative block and why it should not be an absolute contraindication for surgery, as this has important clinical implications.

Figure 5: The bar chart in Figure 5 is effective for showing the magnitude of change. However, it would be strengthened by the inclusion of error bars (e.g., interquartile range for the medians) to visually represent the variance in the data.

Reviewer #3: The authors provided a new concept-Human Disharmony 5 Loop (HDL) as a model that anatomically explains a mix of intractable shoulder pain, persistent 3 impingement/loss of shoulder motion, neck pain, headaches, and distal neuropathy. pectoralis minor release was an effective method to reduce shoulder pain, headaches, 24 concomitant neuropathic lesions and improved shoulder range of motion. This article will provide some suggestion to treat patients with similar symptoms. But there are some questions need to be revised or explained.

1. The authors studied 115 patients in this article, but the patient enrollment period was not mentioned.

2. Some figures (figure 1-4) in this article were similar with the previous paper (reference 10-The Human Disharmony Loop: A Case Series Proposing the Unique Role of the Pectoralis Minor in a Unifying Syndrome of Chronic Pain, Neuropathy, and Weakness, PMID: 40095905). Dr. James M Friedman was the co-author in both articles. I think the authors need explain the two articles' connections and distinctions and add the reference in the figure 1, figure 2, figure 3, and figure 4.

3. In the method part, Patients were evaluated pre-operatively, and at 2, 6, 12, and 24 weeks, so what’s the changing trend of pain scores, clinical neuropathic lesions, rotator cuff impingement 13 signs, shoulder range of motion, and complications at these time points.

4. How about the duration of the 115 patients before operation? If the duration of illness influences the surgical outcome, especially for the neuropathic symptoms?

**Do you want your identity to be public for this peer review?** For information about this choice, including consent withdrawal, please see our Privacy Policy

Reviewer #1: No

Reviewer #2: No

Reviewer #3: No

---

## [Author Response · Author response to Decision Letter 1]

13 Aug 2025

Thank you for taking the time to review our article. In addition to the reviewer comments, we have added the important point that our study gives direct evidence that HDL diagnostic criteria predicts good outcomes with PM release but does not prove the exact accuracy of the HDL model. As with all models, we expect minor changes to be made to the HDL model as more and longer term data is collected. We have made several edits throughout the paper to reflect this. Edit descriptions and rebuttals below.

Reviewer #1:

- I confirm that the article is clearly written, with logical conclusions. Therefore, I accept the publication of this article.

Reviewer #2:

- Study Design and Inconsistency: The most critical issue is a major inconsistency regarding the study design. The abstract states the purpose is to "validate this model prospectively" , whereas the Methods section and main body repeatedly describe the study as a "retrospective chart review". This is a fundamental contradiction that must be corrected throughout the manuscript. The entire framing of the paper depends on it.

- We strongly agree with this statement. This was a retrospective chart review. In our current draft we cannot find any instance where the study is listed as prospective and no edits are highlighted in the manuscript. I have updated the abstract section in the PLOS One submission section.

- Lack of a Control Group: As a retrospective case series, the study lacks a control group. This is a significant limitation. The dramatic improvements could be influenced by the placebo effect, which is known to be substantial in surgical interventions for pain. The authors should more thoroughly discuss this limitation in the Discussion section and acknowledge how it might temper the interpretation of the results. Without a comparison group (e.g., patients meeting HDL criteria treated with conservative therapy alone for the same duration), it is difficult to definitively attribute the outcomes solely to the PM release.

- We agree that this is an inherent limitation to a retrospective case series. We have included this as a limitation to our study (Line 304-309).

- Reliance on the Scratch-Collapse Test: The diagnosis of multiple neuropathic lesions relies on the scratch-collapse test. As the authors rightly note in their limitations, this test is not part of the formal diagnostic criteria for these conditions. While useful as a clinical tool, its reliability and specificity can be debated. This reliance weakens the claims regarding the resolution of specific nerve compressions (e.g., carpal, cubital tunnel). The authors should further contextualize these findings and be more cautious in their conclusions about "curing" these neuropathies.

- We understand this point and agree with your comments regarding the scratch-collapse test. A truly sensitive test for neuropathic lesions does not exist making reliable diagnosis difficult and controversial. We included our definition of neuropathic lesions in the methods and listed it as a limitation. We reworded our discussion (Line 266-7) and limitations (Line 314) to highlight the fact that were are using the scratch-collapse test to measure outcomes.

- The "Double Crush" Phenomenon and Secondary Surgery: The finding that 25% of patients required a secondary neurolysis is clinically very important and adds credibility to the report. However, the discussion could explore this further. Does this high rate of secondary surgery suggest that the HDL model, while valuable, may be incomplete? Or does it position PM release as simply the first critical step in a multi-stage treatment for a more complex regional pathology? Expanding on this would add significant depth to the discussion.

- Thank you for this thoughtful comment. Based on our findings that many distal neurogenic symptoms do improve with PM release we believe that a PM release represents a first step to ‘break the loop’ followed by secondary procedures for persistent isolated symptoms. We have included this in our paper (Line 282-284)

- Statistical Methods Description: In the Methods section, please specify the exact statistical tests used to generate the p-values (e.g., Wilcoxon signed-rank test for pre- vs. post-operative median scores, Chi-squared or Fisher's exact test for categorical variables).

Thank you and we have added this to the Methods section (Lines 124-125)

- Role of Preoperative PM Block: The data shows that 89% of patients who received a PM block reported improvement. This implies 11% had a "negative" block but still proceeded to surgery and, presumably, had good outcomes. The discussion could briefly touch on the potential reasons for a false-negative block and why it should not be an absolute contraindication for surgery, as this has important clinical implications.

- Thank you for this comment. We have added this(Line 300-302)

- Figure 5: The bar chart in Figure 5 is effective for showing the magnitude of change. However, it would be strengthened by the inclusion of error bars (e.g., interquartile range for the medians) to visually represent the variance in the data.

- Thank you. We have added IQR to the relevant variables in figure 5

Reviewer #3:

- The authors studied 115 patients in this article, but the patient enrollment period was not mentioned.

- We first started performing this procedure in 6/2024. IRB exemption for a retrospective chart review of reported variables was obtained on 7/11/24 and 10/14/24 at our respective institutions. A chart review for this paper was carried out on 5/1/25. As this was a chart review, patients were not specifically enrolled for this study.

-

- Some figures (figure 1-4) in this article were similar with the previous paper (reference 10-The Human Disharmony Loop: A Case Series Proposing the Unique Role of the Pectoralis Minor in a Unifying Syndrome of Chronic Pain, Neuropathy, and Weakness, PMID: 40095905). Dr. James M Friedman was the co-author in both articles. I think the authors need explain the two articles' connections and distinctions and add the reference in the figure 1, figure 2, figure 3, and figure 4.

- We did include reference to our first paper in the introduction. We have added further detail to this paper delineating the differences between the two papers (line 65, line 106)

- In the method part, Patients were evaluated pre-operatively, and at 2, 6, 12, and 24 weeks, so what’s the changing trend of pain scores, clinical neuropathic lesions, rotator cuff impingement 13 signs, shoulder range of motion, and complications at these time points.

- We agree that the natural post-operative history for these patients is an interesting topic. The rate of symptom improvement overall is immediate but also dependent on variables such as, but not limited to: BMI, workers comp status and history of prior surgery. Given the complexity of the answer to this question we believe that the post-operative course of patients is beyond the scope of this paper. Future work will focus on many topics we were unable to include in this study.

- How about the duration of the 115 patients before operation? If the duration of illness influences the surgical outcome, especially for the neuropathic symptoms?

- This is an interesting point. Almost all patients had pain for at least 6 months, and many patients had years of pain before treatment. Unfortunately, this is not a variable we recorded for this study. All patients did not respond to at least 2 months of targeted PT as described in our methods. Although we do not feel that this variable was necessary to test our hypothesis, we will include this in future studies.

---

## [Decision Letter · Decision Letter 1]

15 Oct 2025

Validation of The Human Disharmony Loop: Pectoralis minor tenotomy significantly reduces pain and improves function in historically challenging patients who meet reproducible and explicit diagnostic criteria

PONE-D-25-26406R1

Dear Dr. Friedman,

We’re pleased to inform you that your manuscript has been judged scientifically suitable for publication and will be formally accepted for publication once it meets all outstanding technical requirements.

Kind regards,

Emil George Haritinian, M.D, Ph.D.

Academic Editor

PLOS ONE

Additional Editor Comments (optional):

Reviewers' comments:

Reviewer's Responses to Questions

**Comments to the Author**

Reviewer #2: All comments have been addressed

Reviewer #3: All comments have been addressed

2. Is the manuscript technically sound, and do the data support the conclusions?

Reviewer #2: Yes

Reviewer #3: Yes

3. Has the statistical analysis been performed appropriately and rigorously?

Reviewer #2: Yes

Reviewer #3: Yes

4. Have the authors made all data underlying the findings in their manuscript fully available?

Reviewer #2: Yes

Reviewer #3: Yes

5. Is the manuscript presented in an intelligible fashion and written in standard English?

Reviewer #2: Yes

Reviewer #3: Yes

Reviewer #2: Thank you once again for the opportunity to review your manuscript titled “Validation of The Human Disharmony Loop.” I appreciate the thoughtful revisions you have made following the initial round of feedback.

The updated version addresses the key concerns raised previously, including clarification of the study design, acknowledgment of limitations such as the absence of a control group, and refinement of the discussion around diagnostic tools and secondary procedures. These changes have strengthened the manuscript and improved its clarity and scientific rigor.

I particularly commend the way you have contextualized the role of the scratch-collapse test, expanded on the implications of secondary neurolysis, and clarified the diagnostic value of the PM block. The figures and data presentation are also improved, notably with the addition of interquartile ranges.

In summary, I find the revised manuscript well-structured, clearly written, and clinically relevant. I have no further concerns and support its publication.

Thank you again for your valuable contribution.

Reviewer #3: Human Disharmony 29 Loop (HDL) was a model that explained a mix of intractable shoulder pain, persistent

27 impingement/loss of shoulder motion, neck pain, headaches, and distal neuropathy. This article provide a new sight to treat the complex shoulder and upper extremity pain and functional disorder for clinal doctors.

**Do you want your identity to be public for this peer review?** For information about this choice, including consent withdrawal, please see our Privacy Policy

Reviewer #2: No

Reviewer #3: No

---

## [Editor Report · Acceptance letter]

PONE-D-25-26406R1

PLOS ONE

Dear Dr. Friedman,

I'm pleased to inform you that your manuscript has been deemed suitable for publication in PLOS ONE. Congratulations! Your manuscript is now being handed over to our production team.

Kind regards,

on behalf of

Dr. Emil George Haritinian

Academic Editor

PLOS ONE